# An effective hotspot mitigation system for Wireless Sensor Networks using hybridized prairie dog with Genetic Algorithm

**Mohammed Y. Aalsalem** ✱

Farasan Networking Research Laboratory, College of Computer Science & Information Technology, Jazan University, Jazan, Saudi Arabia

* aalsalem.m@jazanu.edu.sa

**Data Availability Statement:** All relevant data are within the manuscript.

**Funding:** The author(s) received no specific funding for this work.

## Abstract

Wireless Sensor Networks (WSNs) consist of small, multifunctional nodes distributed across various locations to monitor and record parameters. These nodes store data and transmit signals for further processing, forming a crucial topic of study. Monitoring the network's status in WSN applications using clustering systems is essential. Collaboration among sensors from various domains enhances the precision of localised information reporting. However, nodes closer to the data sink consume more energy, leading to hotspot challenges. To address these challenges, this research employs clustering and optimised routing techniques. The aggregation of information involves creating clusters, further divided into sub-clusters. Each cluster includes a Cluster Head (CH) or Sensor Nodes (SN) without a CH. Clustering inherently optimises CHs' capabilities, enhances network activity, and establishes a systematic network topology. This model accommodates both multi-hop and single-hop systems. This research focuses on selecting CHs using a Genetic Algorithm (GA), considering various factors. While GA possesses strong exploration capabilities, it requires effective management. This research uses Prairie Dog Optimization (PDO) to overcome this challenge. The proposed Hotspot Mitigated Prairie with Genetic Algorithm (HM-PGA) significantly improves WSN performance, particularly in hotspot avoidance. With HM-PGA, it achieves a network lifetime of **20913 milliseconds** and **310 joules** of remaining energy. Comparative analysis with existing techniques demonstrates the superiority of the proposed approach.

## 1. Introduction

Wireless Sensor Networks (WSNs) are networks of spatially distributed autonomous sensors that monitor physical or environmental conditions, such as temperature, sound, pressure, etc., and cooperatively pass their data through the network to a central location for processing and analysis. A typical WSN usually holds primary blocks necessary to observe, process, and communicate with neighbouring nodes [1]. These blocks enable the users to sense, actuate, and transmit the information according to the requirements of the identified circumstances [2].

**Competing interests:** The authors have declared that no competing interests exist.

The structure of a WSN includes numerous sensor nodes (SNs) that are typically battery-powered and operate wirelessly, making them ideal for deployment in a variety of environments. These SNs not only collect data but also have the capacity to process and route data to other nodes or a base station, which is often referred to as the sink. This functionality enables WSNs to be used in diverse applications ranging from environmental monitoring to security surveillance [3]. The user may be associated with business, medical, civil, or government operations. The conditions may be a framework, biological system, or physical environment. Some WSN applications include surveillance, telemedicine, data collection, and forest fire monitoring. WSNs are typically deployed in remote locations where direct human access is limited. WSNs are particularly suited to remote deployments due to their ability to operate autonomously and their resilience to environmental factors, enabling long-term data collection without human intervention [4]. This deployment strategy is cost-effective, allows monitoring of inaccessible or hazardous areas, and supports environmental research in ecosystems and natural landscapes [5]. These sensor nodes of WSN are battery-powered. Battery power is the primary source of energy for an SN. WSNs are typically deployed in remote locations that humans cannot reach, but it should be noted that WSNs have limited energy [6]. Direct communication between nodes requires more energy. Multi-hop communication uses energy more efficiently as each SN should communicate its sensed data only to its neighbour SN. Typically, most WSNs use multi-hop communication, and many researchers concentrate on the energy efficiency of their neighbour SN [7, 8].

Hotspot issues correlate to ineffective energy consumption (EC), and this research emphasises mitigating the impact of a hotspot in WSN [9]. A "hotspot" in WSN refers to a situation where specific nodes or regions within the network experience higher levels of activity, data transmission, or energy consumption compared to other nodes or regions. This imbalance can lead to issues such as premature node failure, network disruption, and data loss. Hotspot mitigation is a significant issue in WSN research because it balances energy consumption and data traffic across the network, enhancing overall performance and prolonging the network's lifetime.

Generally, WSN employs multi-hop communication instead of direct communication, and hop-by-hop is incorporated for communicating the sensed information to the sink. In this context, a node is nearer to the exchange number of packets than another node in the network [10, 11]. The node near the sink consumes more energy, and they end up using all the energy in advance, which leads the node to die and cut off the network (*Hotspot Problem)*. Hotspot mitigation remains a crucial challenge in the realm of Wireless Sensor Networks (WSNs). Hotspots typically occur in WSNs due to uneven distribution of workload among sensor nodes, leading to rapid energy depletion in certain nodes, thereby reducing network efficiency and lifespan. Some methods have been proposed in the existing literature to lower total Energy Consumption (EC) [12]. Traditional methods for addressing hotspots in WSNs have primarily focused on uniform energy distribution and efficient routing protocols. However, these approaches often fall short in dynamic network environments where node density and data traffic can vary significantly.

This research focuses on lowering WSN hotspot issues. This research uses bio-inspired optimisation to select the Cluster Head (CH) and perfect the routing process [13]. The Genetic Algorithm (GA) is an effective search technique in the multimodal environment and has proved relevant in optimisation-based problems [14]. Based on genetics and evolution, GA can find the best solution. GA can also find a search area's best features [15]. The random distribution of chromosomes achieves GA [16]. The next generation of nodes with various properties is produced when crossover and mutation occur [17]. The GA method was evaluated, and the chromosome that leads to the best solution was given the context of increased

possibilities. Due to its enormous exploration capacity, GA has been used in many fields but cannot exploit the best nodes around the search space [18].

Exploration and exploitation are the two primary skills that evolutionary algorithms develop. Exploration searches for the best solutions in the new search region, whereas exploitation uses existing solutions, and their improvement can increase the Fitness Value (FV) [14, 19]. Prairie Dog Optimization (PDO) is a new metaheuristic approach that draws inspiration from nature and emulates–as the name implies—the behaviours of prairie dogs. The two main optimisation phases are exploration and exploitation, which are conducted using four prairie dog behaviours using the proposed method. The PDO algorithm uses prairie dogs' feeding and burrow-building behaviours to explain such behaviours. Exploration and exploitation should be balanced well in the PDO. The PDO exhibits more robust performance and better capabilities when compared to the findings of other popular population-based metaheuristic algorithms [20].

The food source initiates the burrowing behaviour of PDO, and it explores the whole food source, whereas a new solution or source of food is identified by exploration. The new tunnel is formulated around the new food source and solution space. A distinct sound or alarm is used for communication, which is used in diverse situations [21]. Such a distinctive sound can reveal the existence of a nearby predator or any new food source. PDO's ability helps identify the demands of nutritional food and secures it from predators. In the case of PDO implementation, these two distinct behaviours cause prairie dogs to congregate at a particular position or a good site, where more search (exploitation) is conducted to identify better or nearly ideal solutions [22].

The Prairie Dog Optimization (PDO) algorithm, when integrated with the Genetic Algorithm (GA), addresses specific challenges and enhances the overall performance of the system. PDO is known for its ability to manage local and global searches effectively, which is crucial in overcoming GA's potential limitations in these areas. In the context of the HM-PGA model, PDO assists in fine-tuning the solutions found by the GA. It does so by adjusting parameters or making slight modifications to the chromosome selections, ensuring that the final solutions are not only optimal in terms of energy efficiency but also robust against network changes. This adaptability is critical in dynamic environments like Wireless Sensor Networks, where node energy levels and network conditions can change rapidly. This research considers parameters like Residual Energy, Node Degree, Distance to Base Station, Data Transmission Rate, etc., while choosing the CH, and GA is incorporated into electing CH. The GA lacks exploitation ability, but this is rectified by integrating it with PDO.

This research introduces the Hotspot Mitigated Prairie with Genetic Algorithm (HM-PGA), a novel approach that significantly enhances WSN performance, particularly in hotspot avoidance. Unlike existing methods, HM-PGA employs a hybridised algorithm that not only optimises energy consumption but also intelligently distributes network load, thereby effectively mitigating hotspots. The performance of WSN is enhanced using the proposed Hotspot Mitigated Prairie with Genetic Algorithm (HM-PGA). Mitigating hotspots using HM-PGA enriches the network's performance, and the performance enhancement is investigated using performance metrics. These performance metrics, namely, Network Lifetime (NL), Energy Consumption (EC), reliability, and Packet Delivery Ratio (PDR), are enhanced and relevant to assessing the performance of WSNs. Network Lifetime measures the duration the WSN can operate effectively before the first node depletes its energy, maximising network longevity [23]. Energy Consumption assesses the energy used by nodes, which is crucial for battery-powered WSNs. Reliability ensures accurate data delivery in critical applications, while Packet Delivery Ratio (PDR) measures efficient data transmission. The introduction of HM-PGA marks a significant stride in WSN research, offering a

more resilient and efficient solution to the persistent challenge of hotspot mitigation. This paper aims to detail the methodology behind HM-PGA, its comparative performance advantages over existing hotspot mitigation techniques, and its potential implications for future WSN applications.

This study is structured as follows. The overview of WSN and hotspot issues in WSN are detailed in Section 1, followed by a comprehensive analysis of hotspot mitigation and optimisation approaches in Section 2. The significance and procedures involved in the proposed HM-PGA are described in Section 3, while the simulation outcome of HM-PGA is illustrated in Section 4. Lastly, the HM-PGA is concluded in Section 5, and a recommendation for future refinement is made.

## 2. Related work

The articles [24, 25] suggest an optimised routing method for the WSN-based Internet of Things (IoT) to mitigate the hotspot problem, utilising the Tunicate Swarm Algorithm (TSA) for the selection of Cluster Heads (CHs). TSA optimises cluster-based routing by considering factors such as the network's average energy, the distances between nodes and the sink, load balancing, and each node's energy condition [26]. This results in effective CH selection, which is critical for improving energy efficiency and load balancing across the network [27, 28]. The proposed approach examines two network scenarios: one where the sink is absent, requiring an alternative data aggregation point, and another representing an ideal condition with the sink present to provide optimal solutions for each WSN context.

Authors in [29] proposed a fault-tolerant Distributed Ant Colony Optimization-based Routing (DACOR) protocol for mitigating the hotspot issue in the fog-enabled WSN framework. DACOR employs a unique approach by considering different Fog Nodes (FNs) and uneven clustering-based network models. DACOR minimises energy consumption by dynamically adjusting CHs to FNs based on distance, ensuring a distributed unequal clustering to several FNs [30]. A probabilistic formula for data routing is also developed to prioritise reliability and energy efficiency [31]. Different Fog Nodes (FNs) and uneven clustering-based network models can enhance the network's performance [32]. The suggested model uses less energy because it does not repeatedly cluster the nodes and adjusts CHs to FNs according to distance. A distributed unequal clustering to several FNs is devised to alleviate the hotspot problem. A separate rule is developed to determine the cluster radius based on essential factors that guarantee energy efficiency and balancing factors. A probabilistic formula for data routing that considers reliability and energy efficiency is developed. While previous approaches, such as TSA and DACOR, have made notable contributions to mitigating hotspots, they may still have limitations in terms of scalability, adaptability to different network scenarios, or computational efficiency. These limitations suggest that there is room for improvement in hotspot mitigation strategies.

Authors in [33] used a Political Optimizer-based Unequal Clustering Scheme (POUCS) to mitigate hotspot issues in WSN. The POUCS technique aims to select CHs and find variable cluster sizes. The POUCS method develops a Fitness Function (FF) from various input parameters to reduce EC and increase network longevity. Meanwhile, authors [34, 35] introduced a Certified Network Defender (CND) approach to improve the network's reliability through the proper routing protocol. This CND protocol provides a sequential identification of the next hop SN to create the link between the source and destination. Also, this protocol employs the typical opportunistic routing scheme to traverse the data packet through the channel. The clustering method is not done in CND because it only focuses on selecting the next hop node.

Therefore, the system is easily prone to bottlenecks and the network is depleted entirely on an unbalanced event.

Sensor Protocols for Information via Negotiation (SPIN) [36, 37] is one of the data-centric protocols designed to send the observed data to all the nodes in the network. SPIN protocols offer an efficient data-centric approach in WSNs by negotiating which nodes will transmit data, thereby reducing the unnecessary information propagation that characterises flooding and gossiping methods. However, SPIN does not directly address the issue of energy imbalance across the network, which can lead to hotspots. These hotspots occur when nodes near the sink or frequently communicating nodes deplete their energy faster than their peers, potentially leading to network gaps and decreasing the system's lifespan. Consequently, while SPIN improves data transmission efficiency, its lack of a mechanism for balancing energy consumption leaves room for hotspot-related problems. The proposed HM-PGA approach aims to fill this gap by providing a solution that not only improves data dissemination efficiency but also specifically targets the energy imbalance issue, offering an alternative routing path to prevent the creation of hotspots and extend the network's durability.

One of the most successful energy-efficient protocols designed is the Low-Energy Adaptive Clustering Hierarchy (LEACH) protocol [38, 39]. The primary aim of this protocol is to improve the network's lifespan by preventing redundant information generated by converged nodes. It also minimises the node's EC by employing the clustering technique [40]. PEGASIS is a bi-fold energy-efficient routing protocol [41] that aims to reduce EC and delay in WSN. This network deploys the nodes homogeneously, and their geographical location is available to their neighbours. Also, PEGASIS is capable of operating energy over arbitrary boundaries. This structure uses Code Division Multiple Access (CDMA) transceivers for communication. It is noted that this protocol relies on an ordered tree structure called the PEGASIS chain that forwards observed data to the sink. Although it uses a parallel aggregation concept, it does not deliver a consistent network during bottlenecks [42].

Authors in [43–45] propose a scheduler algorithm for multi-path propagation using Multi-path Transmission Control Protocol (MPTCP), reducing the network's energy consumption. By offline processing the Markov decision, they analyse multiple application schedules, depending on the earlier history of communications and various radio model energy interfaces. Even though this technique attempts to conserve the EC by the network, the unavailability of a hotspot detection protocol is one of its significant disadvantages.

Most current protocols attempt to lower the network's total EC and do not explicitly target reducing the hotspot impact. Although some methods, such as [46, 47], attempt to diminish the hotspot effect, they have not clearly outlined a suitable distribution of nodes and energy to lengthen the network's lifetime and lessen its influence. The motivation for this research comes from this circumstance. There are many bio-inspired approaches available for solving complex and real solutions, for example, the Discrete Rat Swarm Optimization (DRSO), Taguchi-based Cuckoo Search Algorithm (TCSA), and Discrete Penguin Search Algorithm (DPSA). Bio-inspired approaches often face the issues of exploitation and exploration capability, which can be addressed by the combinational technique [48–50]. The GA follows the algorithm parameters, namely initialisation, fitness evaluation, crossover, mutation, and termination, whereas the data is explored and exploited. Likewise, bio-inspired strategies follow different parameters for conducting exploration and exploitation. Hence, this section discusses bio-inspired approaches, identifies a research gap, and makes a simulation comparison using similar bio-inspired approaches. The HM-PGA is a proposal for guiding how the nodes should be distributed throughout the several tiers to reduce the impact of the hotspot problem.

## 3. Proposed methodology

This section elucidates the process adopted to resolve hotspot issues in WSNs by employing a hybrid approach that combines the prairie dog model with a Genetic Algorithm (GA), referred to as the Hotspot Mitigated Prairie with Genetic Algorithm (HM-PGA).

### 3.1 Network model

This research introduces a multi-hop network model utilising the Hotspot Mitigated Prairie with a Genetic Algorithm (HM-PGA), which includes Sensor Nodes (SNs) with an energy harvesting feature and a single Base Station (BS) that provides an endless energy supply. This model supports efficient data routing and energy usage, with SNs functioning as data collectors and routers [51, 52]. The data is routed using BS and sampled using SN, where these SNs act as routers. Data fusion is applied to minimise sent data, and SN's transmission power can vary based on distance. The HM-PGA is signified using a directed graph G = (V, E). The node vertices ($v$) belong to V, and the edge ($u$, $v$) belonging to E of a wireless link among the nodes ($u$, $v$) belongs to V, which allows them to exchange data packets. The proposed approach strategically adjusts SN transmission power based on their distance to reduce energy consumption, a critical factor in mitigating hotspots.

The cluster's energy use for information transmission is proportional to distance $d^2$, and CH to BS is $d^4$. The way to attain an adequate Signal-to-Noise Ratio (SNR) is given in Eqs (1) and (2), which are written here:

$$En_{TX}(k, d) = En_{elect}K + En_{fs}kd^2 \tag{1}$$

$$En_{TX}(k, d) = En_{elect}K + En_{mp}kd^4 \tag{2}$$

Where the message count is shown using '$k$', and the distance is indicated by '$d$'. The energy dispersed per bit to run the receiver or transmitter circuit is $En_{elect}$ (nJ/bit), and the amplifier relies on the distance between the transmitter and receiver is given as $En_{fs}(pJ/bitm^{-2})$, $En_{mp}(pJ/bit \, m^{-2})$. The EC of the receiver is equated in Eq (3):

$$En_{RX}(k) = En_{elect}K \tag{3}$$

The constant value used in the model is $En_{elect}$ = 50 nJ/bit, $En_{fs}$ = 10 pJ/bitm$^{-2}$, $En_{mp}$ = 0.0013 pJ/bitm$^{-4}$. The data fusion network deployed in the research has gathered an '$n$' number of cluster nodes where every node collects '$k$' data bits that are compressed to '$k$' data bits. Fig 1 depicts the radio model, and Table 1 lists its attributes. It details how data is transmitted and received within the network, focusing on the energy consumption aspects. Key components include the number of data bits managed by each node, the distance over which data is transmitted, and energy parameters for operating the circuit and amplifiers.

Node disjoint requirements are sterner than link disjoint necessities. Thus, disjoint node pathways have lower collision probabilities than disjoint link routes. Therefore, the route identification protocol primarily selects the disjoint node pathways as the secondary path. The link disjoint path will be chosen without a node disjoint path. The WSN has two channels with the lengths r and s, where the complexity is determined by comparing all the nodes in the network path to $O(rs)$. For link disjoint, the complexity among two paths is $O((r - 1)(s - 1))$. This complexity can be further diminished based on the deployment of wireless network infrastructure. It is preferable to sort the pathways before the comparison if the addresses of the nodes in wireless networks can be sorted. The mechanism of node disjoint is given in Algorithm 1.

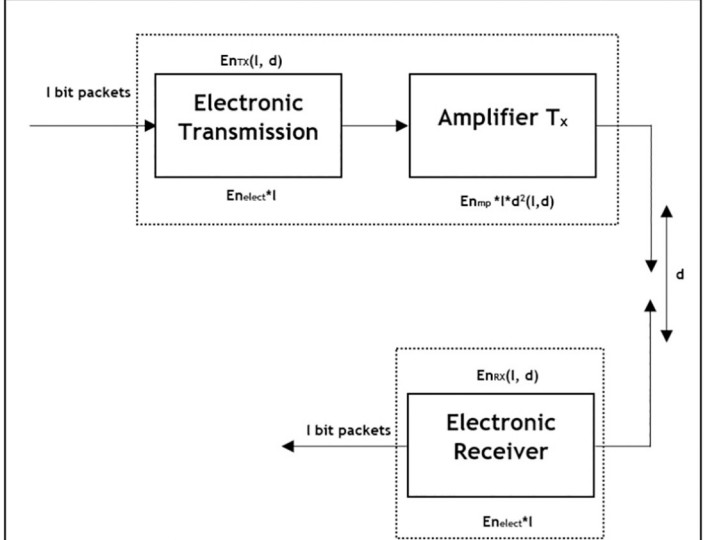

**Fig 1. Radio model.**

```
Algorithm 1: Procedure of Node Disjoint
Input: Length of paths 1 and 2
Step 1: Sorting Path 1 and Path 2
Step 2: Initialisation of l and m = 0
Step 3: while l < path1.length or m < path2.length do
Step 4: if path1[l] = path2[m] then
Step 5: return False
Step 6: else if path1[l] < path2[m] then
Step 7: l++
Step 8: else if path1[l] > path2[m] then
Step 9: m++
Step 10: end if
Step 11: end while
Step 12: return True
```

The cost of EC is the set of data aggregation of $En_D = 5\ nJ/bit$. Radio communication is the primary EC of WSN. The EC model of HM-PGA at node '*n*' is mathematically given in Eq (4):

$$P_n(\tau) = \min(p_n(\tau - 1) + P_{EM,n}(\tau - 1), En_{M,n}) - I(a_n(j)(En_{TX} + En_{RX}) \tag{4}$$

Where the timeslot is indicated using $\tau$, the remaining energy is indicated by $P_n(\tau)$, the node is indicated by '*n*', battery capacity is indicated as $En_{M,n}$, and energy refill in the preceding time-slot is $P_{EH,n}(\tau - 1)$. The largest EC is not allowed to exceed $En_{M,n}$, and the indicator function of WSN is I. The event in node n transmits and receives packets $a_n(j)$. Each node must have a constraint $P_n(\tau) > 0$. The routing protocol is formulated to avoid the closing of nodes. The

**Table 1. Radio model characteristics.**

| Operation | Dissipated Energy |
|---|---|
| Transmitter ($En_{Tx\text{-}choose}$) | 50 *nJ/bit* |
| Receiver ($En_{Rx\text{-}choose}$) | |
| ($En_{Tx\text{-}elect} = En_{Rx\text{-}elect} = En_{elect}$) | |
| Amplifier ($En_{mp}$) | 100 *pJ/bit/m²* |

counter-constrain of WSN is workload maximisation and supporting energy sustainability. The revenue attained by the $j^{th}$ routing packet is R(j) to the BS via WSN, and the main aim is to enhance the whole revenue over the determined time of [0, t]. The total revenue is given using Eq (5):

$$G_t := \sum_{j:jst} R(j)I(aj) \tag{5}$$

where the event $a(j)$ in the $j^{th}$ packet is transmitted and received, and the total count of the packet transferred is T in the time slot is [0:t]. The throughput in a specific time slot is given as '$G_t$'.

The count of SN in WSN is n, and the network is divided into circular tiers where the sink is in the middle region. The count of the node in every tier is $x_i$, where the tier count is shown as $i$. The required energy level to receive data is Rx, and the level of energy needed to transmit data is $Tx$. The EC is equated in Eqs (6) and (7):

$$E_i = \frac{\left[N - \sum_{k=i+1}^{n} xk\right]}{x_i} Rx + \frac{\left[N - \sum_{k=i}^{n} xk\right]}{x_i} Tx \tag{6}$$

$$MAX\{Ei\} = El \tag{7}$$

Where El shows the EC at the sensing region, and a hotspot can occur at the sensing region.

The HM-PGA is deployed in the multi-hop WSN, and the SN is equipped with an additional energy source. The node with high energy plays a significant role in gathering data from the SNs in the network. The composition of energy is discussed in the following section. The normal, intermediate, and advanced node count is formulated in Eqs (8) to (16). $cm$ and $cm_o$ denote respectively the percentage of complex and simple nodes:

$$N_{NRML} = n \times (1 - nm - nm_o) \tag{8}$$

$$N_{INTR} = n \times nm_o \tag{9}$$

$$N_{ADVC} = n \times m \tag{10}$$

$$E_{NRML} = E_o \times (1 - nm - nm_o) \times n \tag{11}$$

$$E_{INTR} = E_o \times (1 + \beta) \times n \times nm_o \tag{12}$$

$$E_{ADVC} = E_o \times (1 + \alpha) \times n \times m \tag{13}$$

$$E_T = E_{ADVC} + E_{INTR} + E_{NRML} \tag{14}$$

$$E_T = E_o \times (1 - nm - nm_o) \times n + E_o \times (1 + \beta) \times n \times nm_o + E_o \times (1 + \alpha) \times n \times m \tag{15}$$

$$E_T = n \times E_o \times (1 + \beta \times nm_o + nm \times \alpha) \tag{16}$$

The normal node is shown by $N_{NRML}$, the intermediate node is shown by NINTR, and NADVC indicates the advanced node. The energy usage of a typical node is shown by $E_{NRML}$, the intermediate node is shown by $E_{INTR}$, and EADVC stands for the advanced node. The total EC of the network is $E_T$. The advanced node is represented by alpha, while the intermediate

node is represented by beta. The energy model of the proposed HM-PGA is given in the above equations.

## 3.2 Hotspot Mitigated Prairie with Genetic Algorithm (HM-PGA)

The Hotspot Mitigated Prairie with Genetic Algorithm (HM-PGA) is a hybrid model that employs Genetic Algorithm (GA) for the initial node alignment and configuration within Wireless Sensor Networks (WSNs). Recognising the inherent limitation of GA in fine-tuning solutions—a process known as exploitation—the model incorporates Prairie Dog Optimization (PDO) to bolster this aspect. PDO is inspired by the behavioural patterns of prairie dogs, particularly their food alarm and anti-predation activities. These are translated into algorithmic procedures to enhance the Cluster Heads (CHs) selection process. In the HM-PGA model, chromosomes represent potential configurations for CHs and are initially generated randomly. Each gene within these chromosomes corresponds to a Sensor Node (SN), with binary values indicating whether the SN is a CH ('1') or not ('0'). Before proceeding to the fitness valuation, a critical validation step ensures that these chromosomes do not violate the current properties and constraints of the SNs, such as energy levels and node distribution, maintaining the network's heterogeneity.

The integration of PDO with GA synergises the explorative strengths of GA with the exploitative precision of PDO, facilitating a balanced approach in searching for optimal network configurations. The PDO's food alarm mechanism acts as a signal to highlight promising configurations, while its anti-predation tactics protect high-quality solutions throughout the evolutionary process. It ensures that the solutions leading to energy-efficient and communication-effective network layouts are discovered, refined, and preserved.

Through this integration, the HM-PGA model systematically mitigates hotspot issues by optimising the distribution of CHs, considering energy consumption patterns, and ensuring effective data transmission across the network. The end goal is to extend the network's lifetime by preventing energy depletion in critical areas, thus maintaining a stable and reliable WSN. The complete methodology, from chromosome initialisation to the final selection of CHs, is visualised in Fig 2, clearly representing the HM-PGA's operational flow and its strategic approach to hotspot mitigation. It depicts the process from chromosome initialisation to the final selection of Cluster Heads (CHs), focusing on setting up network parameters like node counts and energy constraints.

**(a) Initialisation.** The process of initialisation is conducted after the validation of the chromosome. Firstly, the network parameters, namely, node counts, energy constraint, sink count, and location, are considered and responsible for the network's performance. Subsequently, energy parameters are initialised, encompassing adequate energy for data transmission. The procedure of CH selection is given in Algorithm 2.

In the HM-PGA model, the selection of Cluster Heads (CHs) with optimal energy levels is primarily driven by the Genetic Algorithm (GA). The GA operates by evaluating various potential CHs based on their residual energy levels and other network parameters, ensuring that only those with higher energy and better positioning are chosen. This is crucial for prolonging the network lifetime and enhancing energy efficiency. The Prairie Dog Optimization (PDO) further fine-tunes this selection by optimising the spatial distribution and workload among the CHs. It helps in adjusting the network based on dynamic changes, such as varying node energy levels, to maintain optimal energy usage.

The algorithm begins by initialising various parameters, including chromosomes, weighted coefficients, and rates for crossover and mutation. It then generates chromosomes for the population and calculates the Fitness Value (FV) using Prairie Dog Optimization (PDO) and Eqs

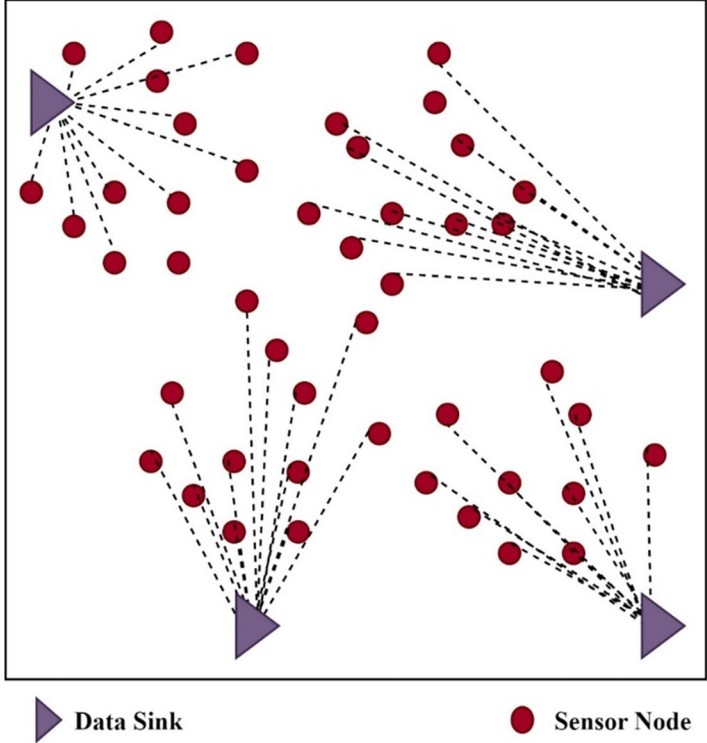

**Fig 2. Network model.**

17–22. The Fitness Function (FF) is computed next. The core of the algorithm involves iterating over generations, where it ranks chromosomes based on fitness, applies genetic operations like crossover and mutation, and then selects the best chromosomes based on an elitism strategy. This process continues until a stopping criterion is met. Integrating GA and PDO, this algorithmic approach ensures efficient CH selection based on fitness values derived from energy levels and other network parameters, leading to energy-efficient routing and data aggregation in Wireless Sensor Networks.

```
Algorithm 2: The proposed HM-PGA for CH Selection
Step 1. Initialisation of chromosome, weighted coefficient, genera-
          tion, crossover, and mutation rate
Step 2. For j = 1:Population Do
Step 3. Chromosome [j]: = getChromosome{set of N}
Step 4. End For
Step 5. For j = 1:Length(Chromosome) Do
Step 6. FV: = getFV(PDO)
Step 7. Compute FV using Eqs 17-22 for every alarm and anti-predation.
Step 8. Computation of FF
Step 9. While Stopping_Criteria_Reached Do
Step 10. Generation = Generation+1
Step 11. Fitness_Chromosome[j] = Rank_Selection(j, F)
Step 12. Choice of Crossover
Step 13. Perform Mutation
Step 14. Chromosome choice using elitism strategy.
Step 15. Repeat
Step 16. End While
Step 17. End For
```

**(b) Fitness Function (FF).**   The process of FF evaluation is a significant aspect of the optimisation technique, and the FV is evaluated by assessing the exploitation ability of the PDO algorithm. The FF is a significant cost function described quantitatively for any feature that must be maximised. The population's weakest members are removed by FF evaluation, leaving only the fittest members. A prairie dog is fittest if its convergence to an ideal point is achieved by moving its fitness rating in that direction. Different FV integrated into the FF is explained in more detail below.

**(c) Food alarm and anti-predation.**   The FV decides any individual SN in the network, and various constraints are considered for formulating the FF. The FPs are programmed to maximise the stability of the network and decrease EC. Three factors are considered while choosing a CH for the network, which are explained below.

The CH uses more energy than the other SN because it has the unique role of gathering, aggregating, and sending the data. CH loses energy more quickly between rounds than typical nodes. As a result, the CH's regular rotation based on stored energy becomes significant. Consequently, the node's remaining energy is essential to consider while constructing the FF. The energy constraint of a node is equated to Eq (17):

$$EC = \sum_{i=1}^{N} \left( \frac{E_{K(i)}}{E_{\max(i)}} + \frac{E_{TH}}{E_T} \right) \tag{17}$$

Where ER indicates Emax, the value of I indicates the remaining energy of ith SN, the maximum range of energy ranges from 1 to N, and ETH stands for the energy threshold. ETH eliminates the node penalisation, and the node with high energy is chosen as CH. The higher value of EC makes the CH high-energy.

The nodes use their energy when communicating with other nodes or a distant sink. The EC increases as the distance between nodes increases during the sending range. In order to create the FF for the selection of CH with the distance factor, the distance is provided in Eqs (18) and (19):

$$DN = \sum_{i=1}^{N} \left( \frac{D_{N-S(i)}}{D_{F(N-S)}} + \frac{1}{D_{AVG\ (N-S)}} \right) \tag{18}$$

$$D_{AVG(N-S)} = \frac{\sum_{i=1}^{N} D_{N-S(i)}}{N} \tag{19}$$

DN estimates the cost of distance incurred for every i[th] SN where the value of $i$ ranges from 1 to N, that is, the entire network count. In Eq 18, $D_{N-S(i)}$ stands for the Euclidean distance of the sink and $i^{th}$ node. A node in the farthest location from the sink is indicated by $D_{F(N-S)}$. The average distance of every node from the sink region is estimated by the Euclidean distance indicated by $D_{AVG(N-S)}$. The higher value of EC ensures that CH has the minimal possible transmission distance.

The efficient use of node energy is greatly influenced by intra-cluster communication. The larger number of nodes near CH assists in saving energy where the usage of energy is enormous. As a result, an alternative node with a greater node density is encompassed by more nodes than any other node in the cluster and is chosen as CH. The node density is decided by

the Eq (20) as follows:

$$ND = \left( \frac{\sum_{i=1}^{N_C} D_{(N-NN)(i)}}{N_C} + \frac{1}{D_{(N-FN)(i)}} \right) \tag{20}$$

where the distance between the neighbor and $i^{th}$ node is shown by $D_{(N-NN)(i)}$. The average Euclidean distance between the farthest and $i^{th}$ node is indicated by $D_{(N-FN)(i)}$. The value of $i$ ranges from 1 to $N_c$, the entire count of a node in the cluster. The average distance of the whole node is indicated by $\sum_{i=1}^{N_C} D_{(N-NN)(i)}$. As seen from Eq 19, $D_{N-NN}$ should be smaller so that the node with the shortest distance to its neighbours is chosen. The value of ND must be maximised for the practical choice of CH that reduces the distance factor and, hence, the nodes' EC.

The SNs are deployed, and they could be closer to one another. There may be some nodes that are not encircled by other nodes since the nodes are distributed randomly. For this reason, the CH node is chosen while considering more nodes around that node, Eq (21):

$$NP = \sum_{i=1}^{n-1} \sum_{j=1}^{n-1} D_{avg_{(i-j)}} \tag{21}$$

where the distance of the $i^{th}$ and the $j^{th}$ node is shown by $D_{avg_{(i-j)}}$.

The node proximity is considered in the above Eq (20) to detect the node closest to the other nodes. Therefore, the FV Node Position (NP) is minimised to choose the node with the highest proximity.

When data transmission commences in the network, the quantity of CHs becomes a significant issue. It is because fewer nodes are left alive when they begin to use their energy. When the count of living nodes decreases and CHs increases, Load Balancing (LB) is necessary, Eq (22):

$$LB = \sum_{i=1}^{n} \frac{Dead_n}{CHN} \tag{22}$$

The terms $CH_N$ and $Dead_n$ in Eq (21) above represent the relative numbers of CHs and dead nodes, respectively. The FV-LB, which decides LB, should be minimised as the number of living nodes declines.

Combining all the weighted values into a single expression may create the FF. The proposed work performance and measure are controlled by the FF provided by Eq (23):

$$Fitness = \frac{1}{(\varphi \times EC) + (\delta \times DN) + (\gamma \times ND) + (\beta \times NP) + (\alpha \times LB)} \tag{23}$$

Where the weighted coefficients are shown by $\varphi$, $\delta$, $\gamma$, $\beta$, and $\alpha$ whereby the values of weighted coefficients are equally distributed. In the HM-PGA model, the Fitness Function (FF) is a crucial determinant of network performance, synthesised by combining all the weighted values into a unified expression as shown in Eq (23). The weighted coefficients are crucial; they emphasise different network attributes such as node energy levels, signal strength, and connectivity. These coefficients are calibrated so their total sum equals one '1', ensuring a balanced influence on the FF as per Eq (24).

$$\varphi + \delta + \gamma + \beta + \alpha = 1 \tag{24}$$

To get the best network performance, PDOs look to minimise FV over processes of evolutionary computation.

**(d) Selection.** The selection procedure is used to improve the quality of the average population in the current generation. The ranking selection and elitist method are used in this study instead of the roulette wheel strategy. In a ranking method, rank is assigned based on the cost value, and the probability of selection is higher; it is more effective than roulette-wheel selection as it does not consider the cost value. The elitist approach is then used to continue with the best-chosen parent node for the following iteration. The previously mentioned node selection procedure is considered. The suitable nodes for choice as CH are listed according to the ranking, and then elitism is used to keep the CH. The same CH is used for the following round until it has sufficient energy, resulting in fewer overhead expenses. The adjacent channel and frequency band are significant factors and considerably impact the interference. WSN is severely affected by interference from the adjacent channels and external interference from other wireless technologies. The performance can be interrupted by interference, and effective data transmission can avoid interference. The interference and other interruptions are handled using effective data transmission and CH selection.

**(e) Crossover.** Crossover is a process in which the mating process results in the recombination of the material parts. The selection procedure based on the PDO node directly changes the crossover's results. A binary genetic operator known as a crossover operator works on two parents. Despite the fact that several crossover methods can be used depending on the situation, in this work, HM-PGA uses single-point crossover, in which the chromosomes are switched after any 1-point, as shown in Table 2. During the crossover operation, one parent chromosome's bit stream is swapped with the other at the crossover location. The two new bitstreams created after the 1-point crossover correspond to two distinct chromosomes and are known as offspring, as stated in Table 2. The crossover rate in network architecture is equivalent to the probability of finding whether there will be a periodic rotation of CHs. The crossover rate, $P_c = 0.6$, is used in the proposed method.

**(f) Mutation.** The next generation of chromosomes will unavoidably inherit the traits of the parent's chromosomes, while the iteration occurs after applying the crossover. Thus, evolutionary optimisation may be lacking in searching among the local solutions. This problem is solved by mutation, which ensures a new gene expression entry into a chromosome. The mutation rate decides how often the mutation will be administered, like the crossover. The mutation rate is either per bit or per chromosome; if it is per bit, there is a 0.01% chance that every bit in a chromosome will become modified. Like this, a 0.001 rate of mutation per bit chromosome assures that each chromosome has a 0.1% chance of being mutated.

In this study, a single chromosome is affected by the mutation administered per bit, as shown in Table 3. For a more significant mutation rate, the eight-bit of offspring 1 is altered or changed from 0 to 1, while offspring 2 remains unchanged. In a network model, the mutation process looks for the best chromosomes by converting CHs into cluster members and vice versa. However, to prevent the number of CHs from growing, the chance of transition from CHs to individuals is kept better. The proposed technique makes use of the mutation rate, $P_m = 0.006$.

**Table 2. Single point's crossover with other points.**

|  | Chromosome | Representation of Bits |
|---|---|---|
| **Before Crossover** | First | 11001 \| 01100000110 |
|  | Second | 10010 \| 10110001110 |
|  | **Offsprings** | **Representation of Bits** |
| **After Crossover** | First | 11001 \| 10110001110 |
|  | Second | 10010 \| 01100000110 |

**Table 3. Process of mutation: An example.**

|  | Chromosome | Representation of Bits |
|---|---|---|
| **Before Crossover** | First | 1100110110001110 |
|  | Second | 1001001100000110 |
|  | **Offsprings** | **Representation of Bits** |
| **After Crossover** | First | 1100110010001110 |
|  | Second | 1001001100000110 |

**(g) Termination.** The termination occurs at a predetermined count of generations. The FV is compared to the preceding iteration throughout the iterations. The chromosomes are revised accordingly, and the best node or chromosome is chosen as a CH. According to the report, selecting a CH with the highest residual energy and the closest communicative distance to the sink is guaranteed by the highest value of the FF.

# 4. Results and discussion

This section evaluates the simulation outcome of the proposed HM-PGA and existing techniques, namely TSA, DACOR, and POUCS, already discussed in section 2. These techniques were selected for comparison because they represent a range of optimisation strategies that address critical operational challenges in Wireless Sensor Networks (WSNs), similar to those tackled by HM-PGA. The performance metrics selected for comparison—End-to-End Delay (EED), Residual Energy (RE), Packet Delivery Ratio (PDR), Network Lifetime (NL), and Network Throughput (NT)—encompass the critical aspects of WSN functionality that are directly impacted by the optimisation techniques employed by HM-PGA and the other algorithms. By analysing these metrics, this paper aims to quantify the enhancements in WSN performance delivered by the HM-PGA and demonstrate its potential for practical applications in various WSN scenarios. The subsequent sections will present and scrutinise the detailed simulation setup and the resulting data.

## 4.1. Network setup

The network model formed in this concept is based on a typical WPAN of the IEEE 802.11 ah standard. The dimensions of the network were 50 $m \times 1000$ $m$. The SN deployment followed a uniform distribution. The spanning distance between the nodes ranged from 5 m to 25 $m$. In this network, 100 SNs were uniformly distributed with an average energy of 0.5 $J$ each. This network was served by one BS or sink node. The observing interval for these sensor nodes was 5 $ms$. Each sensor node could transfer packet sizes up to 1024 bits per second (*bps*). The bandwidth available for this network model was 20 $MHz$. The overall simulation time was 900 s. The application type was user datagram protocol/constant bit rate. The network parameters used for the simulation were referred from literary works and are listed in Table 4.

## 4.2. Simulation discussion

The simulation is investigated for approaches, namely HM-PGA and existing techniques, specifically, Tunicate Swarm Algorithm (TSA) [25], Distributed Ant Colony Optimization-based Routing (DACOR) protocol [29], and Political Optimizer-based Unequal Clustering Scheme (POUCS) [33]. Several key metrics are employed to assess the HM-PGA model's performance comprehensively. End-to-end delay (EED) is crucial for understanding the time efficiency of data transmission, which is important in time-sensitive applications. Residual Energy (RE)

**Table 4. The configuration of a network.**

| Simulation Parameter | Description |
|---|---|
| Area of Distribution | 50*1000 $m$ |
| Node Count | 500 |
| Node Energy at the Initial Stage | 0.5 $J$ |
| Coverage Area | 50 $m$ |
| Simulation Time | 900 $Sec.$ |
| Bandwidth | 20 $MHz$ |
| Packet Size | 1024 $bits$ |

reflects the model's effectiveness in energy conservation, a critical factor for network longevity. Packet Delivery Ratio (PDR) is used to gauge the reliability of data transmission, indicating the success rate of packet delivery. Network Lifetime (NL) measures the duration until network failure, providing insights into the model's ability to balance energy consumption. Finally, Network Throughput (NT) assesses the data handling capacity of the network, a vital aspect of overall network performance. These metrics together offer a robust evaluation of the network's efficiency, reliability, and operational effectiveness.

As mentioned earlier, the investigation of the algorithms' performance compared to other state-of-the-art algorithms has been given in detail below.

**(a) End-to-End Delay (EED).** Some possible reasons for EED are the best route selection, queue length, and communication period. The following Eq (25) is used to decide the EED:

$$EED = \frac{\sum (s_i - s_{i-1})}{D} \tag{25}$$

Where $s_i$ denotes the data packet reaching the destination node, si-1 represents when the source node transmits the first data packet, and D stands for the total number of data packets transmitted. The simulation study of EED according to the number of nodes is shown in Fig 3 and Table 5.

Fig 3 presents a visual comparison of the HM-PGA algorithm's performance against other methods in minimising End-to-End Delay in WSNs. It is seen that the proposed HM-PGA protocol experiences minimum EED when compared to the available protocols. The proposed method reaches a delay of 90, 211, 383, 458, and 591 milliseconds ($ms$) for the diverse count of nodes. The proposed approach achieves a minimal EED when compared to existing techniques.

**(b) Residual Energy (RE).** The parameter RE finds the sensor nodes in the routing path have higher remaining energy. The SN with maximum RE can participate in the communication for a longer duration. Therefore, a system with a high RE is considered an energy-efficient WSN. The analysis of RE for the proposed HM-PGA and existing methods, namely TSA, DACOR, and POUCS protocols, are depicted in Fig 4 and Table 6. The HM-PGA protocol proved the maximum RE when compared to the other protocols. Fig 4 demonstrates the effectiveness of HM-PGA in maintaining higher residual energy in the sensor nodes within the WSN, which is a key indicator of energy efficiency. The graph or chart in Fig 4 probably illustrates how the HM-PGA protocol ensures that sensor nodes have a higher remaining energy over time compared to other methods, thus allowing for longer communication durations and more efficient network operation.

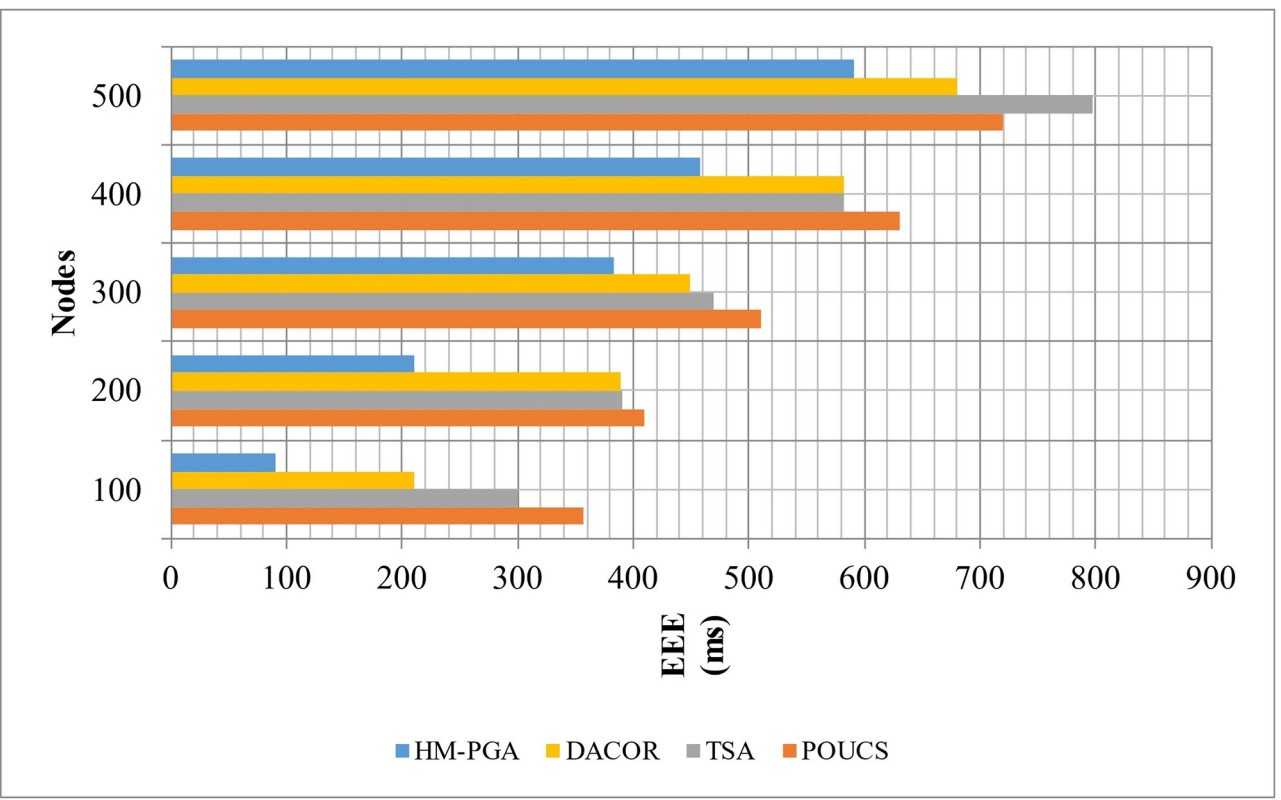

**Fig 3. Comparative graphical analysis of EED.**

In the existing DACOR technique, the energy level at the simulation time of 100 *s* was 190 *mJ*. However, the efficient clustering and optimisation routing algorithm in HM-PGA achieved higher residual energy; thus, its energy level at 100 s is 310 *mJ*.

**(c) Packet Delivery Ratio (PDR).** PDR is a proportion of the packets received at the terminal node to those transmitted from the source node. The following Eq (26) is used to obtain the value of PDR. As Table 7 and the graphical representation in Fig 5 illustrate, while parallel connections are augmented, the proposed HM-PGA delivers high PDR compared to the other protocols.

$$PDR = \frac{Total\_Packets\_Received}{Total\_Packets\_Sent} \qquad (26)$$

**Table 5. Comparative analysis of EED.**

| Nodes | POUCS | TSA | DACOR | HM-PGA |
|---|---|---|---|---|
| 100 | 357 | 300 | 210 | 90 |
| 200 | 410 | 390 | 389 | 211 |
| 300 | 510 | 470 | 449 | 383 |
| 400 | 630 | 582 | 582 | 458 |
| 500 | 720 | 798 | 681 | 591 |

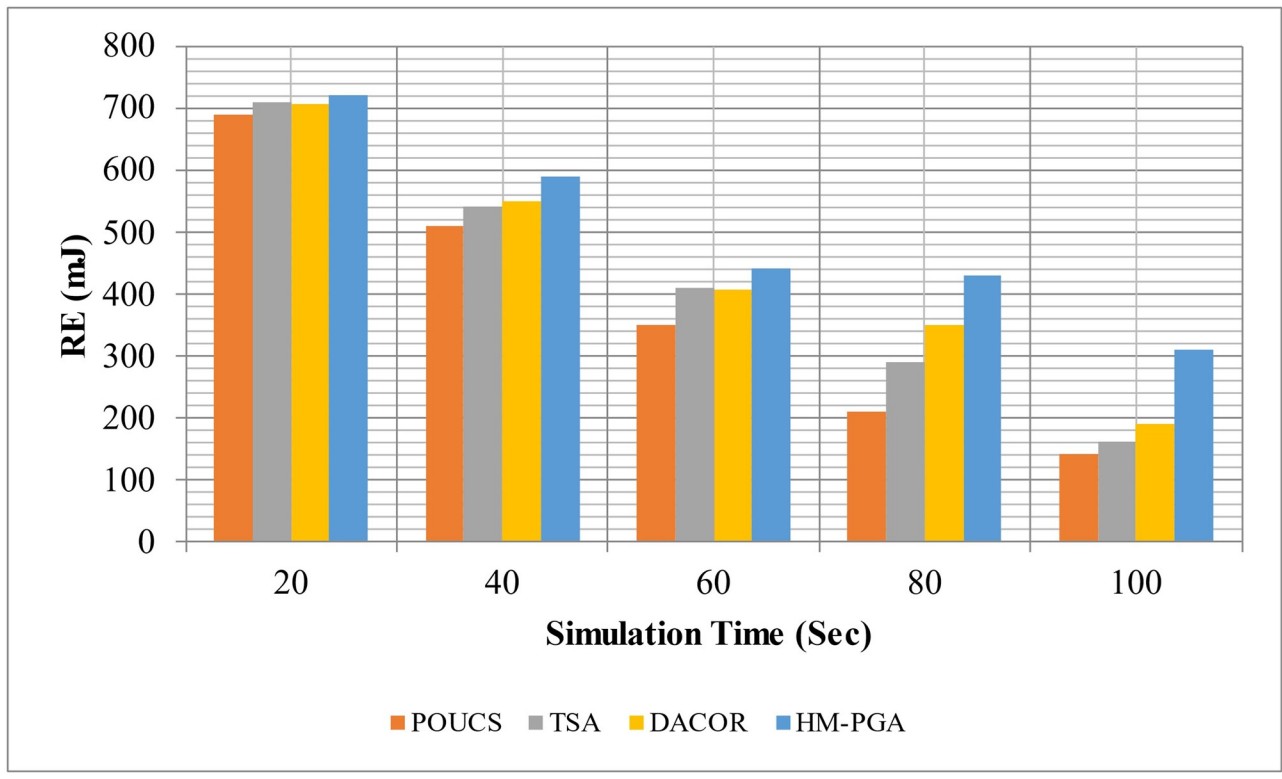

**Fig 4. Graphical evaluation concerning RE.**

The observation shows that the proposed HM-PGA achieved PDR of 93%, 89%, 87%, 83%, and 81% for the respective node counts of 100, 200, 300, 400, and 500. The PDR of HM-PGA is higher compared to existing techniques. Effective mitigation and robust routing have achieved higher PDR in the proposed HM-PGA.

**Table 6. Evaluation in relation to RE.**

| Simulation Time (s) | POUCS | TSA | DACOR | HM-PGA |
|---|---|---|---|---|
| 20 | 690 | 710 | 705 | 720 |
| 40 | 510 | 540 | 550 | 590 |
| 60 | 350 | 410 | 405 | 440 |
| 80 | 210 | 290 | 350 | 430 |
| 100 | 140 | 160 | 190 | 310 |

**Table 7. Comparative analysis of the PDR.**

| Connection (Count) | POUCS | TSA | DACOR | HM-PGA |
|---|---|---|---|---|
| 2 | 79 | 81 | 85 | 93 |
| 4 | 72 | 76 | 84 | 89 |
| 6 | 71 | 75 | 81 | 87 |
| 8 | 70 | 74 | 79 | 83 |
| 10 | 69 | 72 | 76 | 81 |

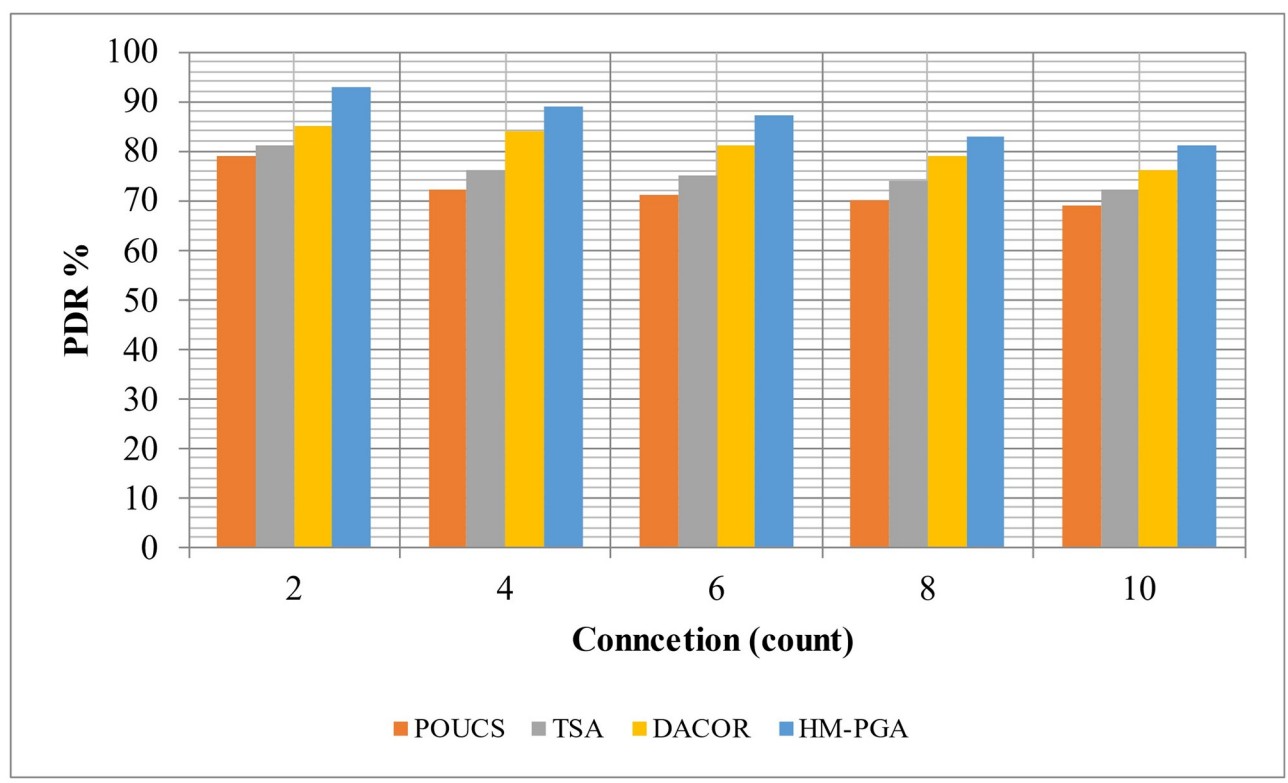

**Fig 5. Graphical analysis of PDR.**

**(d) Network Lifetime (NL).** An essential performance measure for WSNs is NL, which is calculated as the period until the first sensor's energy runs out. In conventional WSNs, every SN is set up to transmit data gathered to the sink via multi-hop transmission. It represents the period during which a WSN would be completely functional. The outcome of NL for the proposed and existing method is given in Table 8 and Fig 6.

The proposed HM-PGA protocol experiences the highest Network Lifetime compared to the available protocols. The proposed method reaches NL of 27627, 24781, 23011, 21098, and 20913 *ms* for the diverse count of nodes. The proposed method achieves the highest NL better than existing techniques. Effective mitigation and robust routing have enabled the proposed HM-PGA to outperform the existing methods.

**(e) Network Throughput (NT).** NT is the ratio of the proportion of data packets bought at the terminal node to the data packets sent by the original node. Table 9 and Fig 7 show the NT analysis for the MH-PGA protocol and the comparison with other available techniques.

**Table 8. Comparative analysis of the NL.**

| Nodes | POUCS | TSA | DACOR | HM-PGA |
|---|---|---|---|---|
| 100 | 19087 | 20671 | 21987 | 27627 |
| 200 | 18972 | 19911 | 20981 | 24781 |
| 300 | 17827 | 18761 | 19078 | 23011 |
| 400 | 16761 | 17565 | 17981 | 21098 |
| 500 | 15861 | 16872 | 16091 | 20913 |

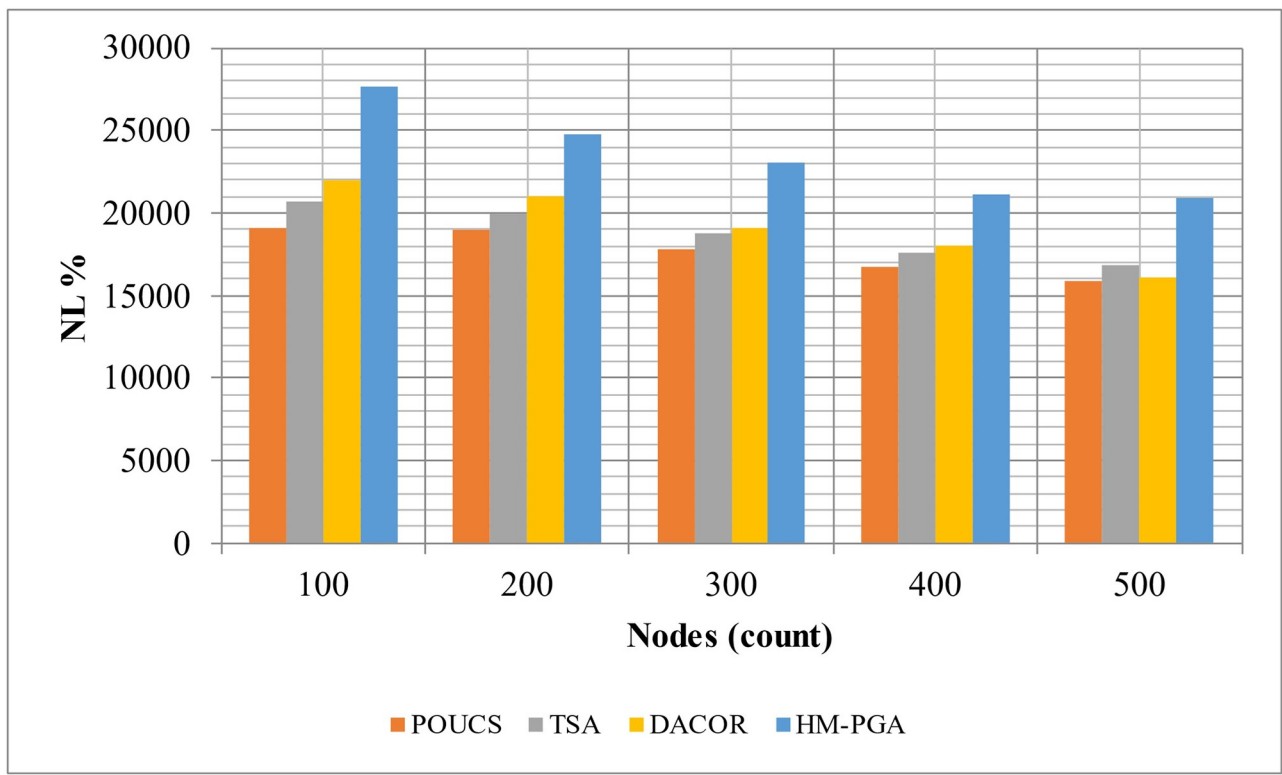

**Fig 6. Graphical performance comparison of the NL.**

The following Eq (27) depicts the measurement of NT:

$$NT = \frac{d}{s} \times 100 \qquad (27)$$

Where $d$ is the number of data packets the destination node receives, and 's' is the number of data packets sent by the source node.

Fig 7 presents a graphical comparison of the NT (in bits per second, bps) achieved by the proposed HM-PGA algorithms compared to other existing methods like TSA, DACOR, and POUCS over different simulation times. The observation shows that the proposed HM-PGA achieved PDR of 3891, 3781, 3617, 3516, and 3481 *bps* for the respective simulation times of 20, 40, 60, 80, and 100. The NT of HM-PGA is higher compared to that of existing methods. The data transmission is compared for the TSA, DACOR, POUCS, and proposed HM-PGA algorithms. The proposed approach is based on the Bio-Inspired technique, so the

**Table 9. Comparison of NT.**

| Simulation Time (s) | POUCS | TSA | DACOR | HM-PGA |
|---|---|---|---|---|
| 20 | 3399 | 3461 | 3672 | 3891 |
| 40 | 3267 | 3387 | 3514 | 3781 |
| 60 | 3156 | 3298 | 3311 | 3617 |
| 80 | 3071 | 3145 | 3298 | 3516 |
| 100 | 2871 | 2908 | 2987 | 3481 |

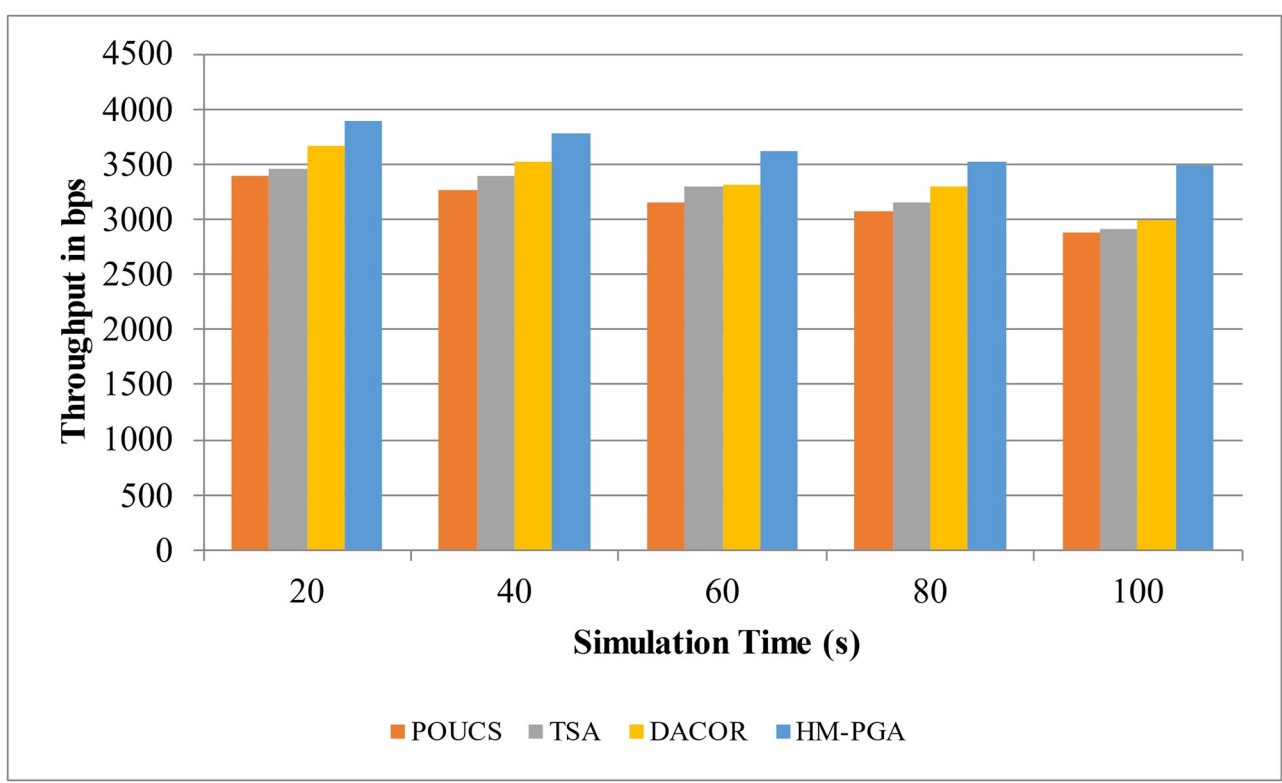

**Fig 7. Results are compared to NT.**

comparison is accomplished among different Bio-Inspired methods. The effectiveness of the simulation is analysed using the performance metrics, namely End-to-End Delay (EED), Residual Energy (RE), Packet Delivery Ratio (PDR), Network Lifetime (NL), and Network Throughput (NT), whereby the comparison is made for a different number of nodes from 100 to 500. For node 500, the EED is 129 ms, 207 ms, and 90 ms, which are minimal compared to existing techniques. The HM-PGA attains the highest PDR of 81% and NL of 20913 ms.

**(f) Sensitivity Study (ST).** This study conducted a sensitivity study for a controlled dataset and limited sample for HM-PGA model, presented in Table 10.

- Population Size: 50, 100, 150

- Mutation Rate: 0.01, 0.05, 0.1

- Node Density: Low (0.1 nodes/unit$^2$), Medium (0.5 nodes/unit$^2$), High (1.0 nodes/unit$^2$)

This table provide a sample result on some for sensitivity analysis. With increasing Population Size, there is a clear trend of improved Network Lifetime (NL) and reduced End-to-End Delay (EED), suggesting that larger populations can enhance network longevity and data transmission efficiency. Moreover, the impact of Mutation Rate on NL is positive across different Population Sizes, indicating that introducing more genetic diversity through higher mutation rates can extend network lifetime. However, the relationship between Mutation Rate and EED is more nuanced, with exceptions observed, particularly in Low Node Density scenarios.

**Table 10. Sensitivity analysis of HM-PGA model.**

| Population Size | Mutation Rate | Node Density | NL (Rounds) | EED (ms) |
|---|---|---|---|---|
| 50 | 0.01 | Low | 1520 | 118 |
| 50 | 0.01 | Medium | 1480 | 112 |
| 50 | 0.01 | High | 1420 | 108 |
| 50 | 0.02 | Low | 1570 | 114 |
| 50 | 0.02 | Medium | 1510 | 109 |
| 50 | 0.02 | High | 1450 | 104 |
| 50 | 0.05 | Low | 1600 | 110 |
| 50 | 0.05 | Medium | 1540 | 106 |
| 50 | 0.05 | High | 1480 | 102 |
| 50 | 0.1 | Low | 1550 | 116 |
| 50 | 0.1 | Medium | 1490 | 111 |
| 50 | 0.1 | High | 1430 | 107 |
| 100 | 0.01 | Low | 1620 | 96 |
| 100 | 0.01 | Medium | 1580 | 92 |
| 100 | 0.01 | High | 1520 | 88 |
| 100 | 0.05 | Low | 1670 | 91 |
| 100 | 0.05 | Medium | 1630 | 87 |
| 100 | 0.05 | High | 1570 | 83 |
| 100 | 0.1 | Low | 1610 | 94 |
| 100 | 0.1 | Medium | 1570 | 90 |
| 100 | 0.1 | High | 1510 | 86 |
| 150 | 0.01 | Low | 1680 | 79 |
| 150 | 0.01 | Medium | 1640 | 75 |
| 150 | 0.01 | High | 1580 | 71 |
| 150 | 0.05 | Low | 1720 | 76 |
| 150 | 0.05 | Medium | 1680 | 72 |
| 150 | 0.05 | High | 1620 | 68 |
| 150 | 0.1 | Low | 1660 | 78 |
| 150 | 0.1 | Medium | 1620 | 74 |
| 150 | 0.1 | High | 1560 | 70 |

Node Density plays a crucial role, showing a trade-off between network lifetime and data transmission efficiency, as denser networks exhibit lower EED but shorter NL. Overall, these findings demonstrate the robustness of the HM-PGA model, which can adapt to various conditions, optimizing network performance based on specific objectives, whether it is prioritizing network longevity or low-latency data transmission.

Based on these findings, it can be stated here that HM-PGA outperforms the existing techniques. HM-PGA's simulation performance suggests its superiority in key areas compared to TSA, DACOR, and POUCS, primarily due to its refined optimisation strategies. Its lower End-to-End Delay is attributed to efficient route selection from integrated PDO mechanisms, while higher Residual Energy results from its dynamic energy management. Improved Packet Delivery Ratio reflects HM-PGA's robust routing protocols, and extended Network Lifetime is achieved through balanced load distribution. Lastly, enhanced Network Throughput is likely due to optimised data transmission paths. These advantages demonstrate HM-PGA's effective utilisation of genetic algorithms and prairie dog behaviours in network optimisation.

## 5. Conclusion

HM-PGA has proven its effectiveness in improving the performance of WSNs. Extensive experimentation has quantified the impressive achievements of HM-PGA, including an exceptional network lifetime of 20,913 milliseconds, surpassing existing state-of-the-art methods. Packet Delivery Ratio (PDR) has notably increased to 81%, while Network Throughput (NT) has also been enhanced, with values reaching as high as 3891 bps during simulations. Furthermore, this approach has significantly reduced Energy Consumption (EC) while extending the network's operational lifespan. These quantifiable results highlight HM-PGA's potential as a robust solution for hotspot mitigation in WSNs, offering promising outcomes for various applications and scenarios. In addition to these achievements, the study acknowledges the challenges faced, such as scalability in diverse network configurations, and invites future research to address these. The broader implications of HM-PGA's performance in WSNs open avenues for its application in more complex and varied environments, encouraging continued innovation in this field.

## 6. Future work

Future research directions aim to refine and optimise the HM-PGA algorithm further by exploring advanced optimisation techniques and innovative algorithms for enhanced hotspot mitigation. This includes the development of adaptive strategies capable of dynamically adjusting to real-time network conditions, promising even more efficient hotspot mitigation. In addition, this research will investigate the integration of HM-PGA with cutting-edge technologies such as the Internet of Things (IoT) and Artificial Intelligence (AI) [53], aiming to unlock new efficiencies and capabilities in Wireless Sensor Networks (WSNs) for applications in environmental monitoring, smart cities, and healthcare [54]. Insights from this study also have broader implications for WSN research, including the development of energy-efficient routing protocols, dynamic sensor deployment strategies, robust security mechanisms, and cross-layer network optimization. These aspects are vital for practical, real-world implementations, and their exploration will enhance network performance and reliability. The continuous advancement of WSN technologies will be driven by these ongoing research efforts, focusing on critical issues like energy efficiency, network reliability, and security in the expanding realm of WSN applications. Further, to explore adaptive mechanisms that dynamically adjust to varying network conditions, enhancing the model's robustness and effectiveness in diverse environments can be a future research direction. Incorporating adaptive mechanisms in the HM-PGA algorithm allows for dynamic adjustments to varying network conditions, significantly enhancing its robustness and effectiveness. This adaptability ensures the algorithm remains efficient even as network topologies, node densities, and energy levels change, leading to improved hotspot mitigation and overall network performance. These efforts will be helpful to broaden its applicability across various sectors including environmental monitoring, smart cities, and healthcare.

## Author Contributions

**Conceptualization:** Mohammed Y. Aalsalem.

**Formal analysis:** Mohammed Y. Aalsalem.

**Funding acquisition:** Mohammed Y. Aalsalem.

**Investigation:** Mohammed Y. Aalsalem.

**Methodology:** Mohammed Y. Aalsalem.

**Validation:** Mohammed Y. Aalsalem.

**Visualization:** Mohammed Y. Aalsalem.

**Writing – original draft:** Mohammed Y. Aalsalem.

**Writing – review & editing:** Mohammed Y. Aalsalem.

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
