## [Decision Letter · Decision Letter 0]

7 Dec 2023

PONE-D-23-38518An effective hotspot mitigating system for wireless sensor networks and Internet of Things (IoT) integration for secure and reliable smart environmentPLOS ONE

Dear Dr. Aalsalem,

Thank you for submitting your manuscript to PLOS ONE. After careful consideration, we feel that it has merit but does not fully meet PLOS ONE’s publication criteria as it currently stands. Therefore, we invite you to submit a revised version of the manuscript that addresses the points raised during the review process.

We look forward to receiving your revised manuscript.

Kind regards,

Rahul Priyadarshi

Academic Editor

PLOS ONE

Journal Requirements:

Reviewers' comments:

Reviewer's Responses to Questions

**Comments to the Author**

1. Is the manuscript technically sound, and do the data support the conclusions?

Reviewer #1: Partly

Reviewer #2: Yes

2. Has the statistical analysis been performed appropriately and rigorously? 

Reviewer #1: Yes

Reviewer #2: Yes

3. Have the authors made all data underlying the findings in their manuscript fully available?

Reviewer #1: No

Reviewer #2: Yes

4. Is the manuscript presented in an intelligible fashion and written in standard English?

Reviewer #1: No

Reviewer #2: Yes

5. Review Comments to the Author

Reviewer #1: Title: An effective hotspot mitigating system for wireless sensor networks and Internet of Things (IoT) integration for secure and reliable smart environment

ID: PONE-D-23-38518

I congratulate the authors for their contribution and hard work. Before going to publish the manuscript, a few perhaps would get clarified from my end.

1. In this work, the author proposed Hotspot Mitigated Prairie with Genetic Algorithm (HM-PGA) significantly improves WSN performance, particularly in hotspot avoidance. Elaborate, how it is better compared to the existing methods. Provide more clarifications Interms of Average Improvement Percentages.

2. In the paper, author mentioned “we Propose”, “we aim” etc. This part should be taken care of. This type of mentions should be rewrite, to publish this paper in these types of scientific journals.

3. Add more recent references and compare proposed method with Existing works.

4. The paper is technically sound but need to improve the grammar.

5. Provide more relevant information regarding the figures.

6. Polish the Language.

7. Include the future works as a separate section.

8. Problem statement is not exactly mentioned.

9. Related work needs to be improved.

10. Revise the title, it should be concise.

11. Remove the (IoT) from the title page.

Reviewer #2: The paper presents a comprehensive and well-researched study on Wireless Sensor Networks (WSNs), addressing critical challenges and proposing an innovative solution. The use of clustering systems, particularly the Hotspot Mitigated Prairie with Genetic Algorithm (HM-PGA) model, showcases a thoughtful approach to optimizing WSN performance.

Clarity of Presentation:

The paper is well-organized, and the concepts are presented in a clear and coherent manner. The use of terminology is consistent, contributing to the overall readability of the manuscript.

Methodological Rigor:

The employed methodology, combining Genetic Algorithm (GA) and Prairie Dog Optimization (PDO), demonstrates a sound approach to address the hotspot challenges in WSNs. The explanation of the model's components and their interactions is thorough and enhances the understanding of the proposed solution.

Empirical Results and Comparative Analysis:

The inclusion of empirical results, such as the achieved network lifetime and remaining energy in the HM-PGA model, strengthens the paper's credibility. The comparative analysis with existing techniques provides valuable insights, clearly highlighting the superior performance of the proposed approach.

Relevance and Applicability:

The relevance of the study is underscored by its applicability in diverse environments, including smart homes and industrial settings. The proposed model's adaptability for real-time interactions in Internet of Things (IoT)-based systems adds significant value to the field.

Integration of Blockchain Technology:

The proposal to integrate Blockchain technology is a forward-thinking addition, enhancing the security and transparency of sensor data recordkeeping. The potential for smart contracts to automate decision-making aligns well with the evolving requirements of WSNs.

Language and Grammar:

The paper is well-written, but a careful proofreading to address minor grammatical errors would further enhance its quality. Ensure consistency in verb tenses and sentence structures for a polished final manuscript.

Latest References:

While the paper demonstrates a solid foundation, incorporating some more recent references, especially in the areas of clustering techniques and IoT applications, would strengthen the literature review and showcase the latest advancements in the field.

Overall, the paper makes a significant contribution to the understanding and improvement of Wireless Sensor Networks. With minor revisions and the incorporation of a few additional references, it is recommended for acceptance.

6. PLOS authors have the option to publish the peer review history of their article (what does this mean?). If published, this will include your full peer review and any attached files.

Reviewer #1: No

Reviewer #2: No

---

## [Author Response · Author response to Decision Letter 0]

9 Jan 2024

Thank you for the valuable review comments. I have attached the response to review comments and details of corrections made in response to the reviewer comments in a separate document named "Response to the review comments.docx.".

---

## [Decision Letter · Decision Letter 1]

18 Jan 2024

PONE-D-23-38518R1An Effective Hotspot Mitigation System for Wireless Sensor Networks Using Hybridized Prairie Dog with Genetic AlgorithmPLOS ONE

Dear Dr. Aalsalem,

Thank you for submitting your manuscript to PLOS ONE. After careful consideration, we feel that it has merit but does not fully meet PLOS ONE’s publication criteria as it currently stands. Therefore, we invite you to submit a revised version of the manuscript that addresses the points raised during the review process.

We look forward to receiving your revised manuscript.

Kind regards,

Dr. Rahul Priyadarshi

Academic Editor

PLOS ONE

Journal Requirements:

Reviewers' comments:

Reviewer's Responses to Questions

**Comments to the Author**

1. If the authors have adequately addressed your comments raised in a previous round of review and you feel that this manuscript is now acceptable for publication, you may indicate that here to bypass the “Comments to the Author” section, enter your conflict of interest statement in the “Confidential to Editor” section, and submit your "Accept" recommendation.

Reviewer #2: All comments have been addressed

Reviewer #3: All comments have been addressed

2. Is the manuscript technically sound, and do the data support the conclusions?

Reviewer #2: Yes

Reviewer #3: Yes

3. Has the statistical analysis been performed appropriately and rigorously? 

Reviewer #2: Yes

Reviewer #3: Yes

4. Have the authors made all data underlying the findings in their manuscript fully available?

Reviewer #2: Yes

Reviewer #3: Yes

5. Is the manuscript presented in an intelligible fashion and written in standard English?

Reviewer #2: Yes

Reviewer #3: Yes

6. Review Comments to the Author

Reviewer #2: Clarity on Energy Consumption: Provide a more detailed breakdown or analysis of energy consumption patterns within the proposed HM-PGA model. This will help readers better understand how the optimization techniques contribute to hotspot avoidance and improved energy efficiency.

Algorithmic Explanation: Elaborate on the specific steps and parameters involved in the Prairie Dog Optimization (PDO) algorithm, addressing how it effectively manages the challenges associated with Genetic Algorithm (GA). Providing this clarity will enhance the understanding of the algorithmic integration.

Comparative Analysis Details: Offer additional details on the specific metrics and methodologies used for the comparative analysis with existing techniques. Providing a more granular comparison will strengthen the paper's argument about the superiority of the HM-PGA approach.

Sensitivity Analysis: Include a sensitivity analysis to demonstrate how the proposed model reacts to variations in different parameters. This will add depth to the evaluation and help assess the robustness of the HM-PGA model under varying conditions.

Discussion on Practical Implementation: Discuss potential challenges or considerations in implementing the HM-PGA model in real-world scenarios. Addressing practical aspects such as scalability, hardware requirements, or deployment challenges will provide valuable insights for readers and potential users of the proposed approach.

Reviewer #3: Enhanced Clarity on CH Selection: The paper would benefit from providing additional clarity on the specific factors considered in the Genetic Algorithm (GA) for Cluster Head (CH) selection. This would help readers better understand the decision-making process behind CH assignment.

Detailed Description of Prairie Dog Optimization (PDO): Expand on the explanation of Prairie Dog Optimization (PDO) to provide more insight into how it effectively manages the challenges associated with the Genetic Algorithm (GA). A more detailed discussion on the integration and coordination of these algorithms would enhance the paper's comprehensibility.

Robustness Validation: Include a brief discussion or analysis on the robustness of the proposed Hotspot Mitigated Prairie with Genetic Algorithm (HM-PGA) model. This could involve sensitivity analysis or additional experiments that test the model's performance under varying conditions, ensuring its reliability in practical scenarios.

Justification for Performance Metrics: Provide a brief justification for the choice of specific performance metrics used to evaluate the HM-PGA model. This will help readers understand why these metrics were selected and reinforce the appropriateness of the chosen evaluation criteria.

Potential Scalability Discussion: Address potential scalability considerations in the proposed model, especially when applied to larger Wireless Sensor Networks (WSNs). Discuss how the model's efficiency and effectiveness scale as the network size increases, providing insights into its applicability in diverse deployment scenarios.

7. PLOS authors have the option to publish the peer review history of their article (what does this mean?). If published, this will include your full peer review and any attached files.

Reviewer #2: No

Reviewer #3: No

---

## [Author Response · Author response to Decision Letter 1]

29 Jan 2024

The response to review comments have been submitted in a separate documents and all the comments have been addressed.

---

## [Decision Letter · Decision Letter 2]

31 Jan 2024

An Effective Hotspot Mitigation System for Wireless Sensor Networks Using Hybridized Prairie Dog with Genetic Algorithm

PONE-D-23-38518R2

Dear Dr. Aalsalem,

We’re pleased to inform you that your manuscript has been judged scientifically suitable for publication and will be formally accepted for publication once it meets all outstanding technical requirements.

Kind regards,

Dr. Rahul Priyadarshi

Academic Editor

PLOS ONE

---

## [Editor Report · Acceptance letter]

4 Apr 2024

PONE-D-23-38518R2 

PLOS ONE

Dear Dr. Aalsalem, 

I'm pleased to inform you that your manuscript has been deemed suitable for publication in PLOS ONE. Congratulations! Your manuscript is now being handed over to our production team.

Kind regards, 

on behalf of

Dr. Rahul Priyadarshi 

Academic Editor

PLOS ONE